# Throwbacks that move us: The dance-inducing power of nostalgic songs

**Riya K. Sidhu**[1,2]*, **Diana M. Urian**[3], **Hong Zheng**[3], **Jessica A. Grahn**[1,2]

**1** Department of Psychology, Western University, London, Ontario, Canada, **2** Centre for Brain and Mind, Western University, London, Ontario, Canada, **3** Schulich Medicine & Dentistry, Western University, London, Ontario, Canada

* rsidhu66@uwo.ca

## Abstract

The urge to move to music, often referred to as groove, is influenced by various factors, including familiarity with the music. The influence of nostalgia, which involves familiarity but also includes pleasant, sad, and wistful emotions, remains largely unexplored. Here we investigate the impact of both familiarity and nostalgia on the desire to tap, move, and dance along to music. To evoke nostalgia, we selected popular songs from the participants' adolescent years. More recent songs served as a low-nostalgia but familiar control. Participants completed an online experiment, rating songs based on their desire for three different movement types (tap, move, and dance), as well as enjoyment, familiarity, and nostalgia. Nostalgic songs elicited higher desire to move than familiar songs across all three movement categories. Additionally, both familiarity and nostalgia predicted move and tap ratings, but only nostalgia emerged as a predictor for dance ratings. Our results suggest a distinctive role for nostalgia, beyond the influence of familiarity, in motivating the desire to dance.

## Introduction

Imagine you are at a friend's wedding reception when a long-forgotten nostalgic song makes you want to get up and dance. Music 'throwbacks' are frequently played at social events to encourage people onto the dance floor. This pleasurable desire to move to music is often referred to as 'groove' [1,2]. Several musical elements affect our experience of groove, such as beat salience, syncopation, and rhythmic complexity [3–5]. Individual factors also influence groove perception, such as genre preference, familiarity, and musical training [6]. One factor that has not been studied, despite its anecdotal relevance, is nostalgia. The current study investigates the effects of nostalgia on groove perception.

**Data availability statement:** The data underlying the results presented in the study are available at https://osf.io/vf3gm/files/osfstorage/67e089c018f1985e1e053eab.

**Funding:** This research was supported by an NSERC Discovery Grant (RGPIN-2016-05834) and NSERC Steacie Fellowship (566202-2021-SMFSU) to JAG. As well as an NSERC Postgraduate Scholarship-Doctoral to RKS. The funding source was not involved in study design, data collection, analysis, interpretation, or manuscript preparation.

**Competing interests:** The authors have declared that no competing interests exist.

## Familiarity as a groove-inducing factor

Among various predictors of groove, familiarity has been identified as one of the most influential factors [1,2,6,7]. This may result from its facilitation of rewarding behaviours and its effect on enjoyment. For example, familiarity enhances our ability to anticipate a song's rhythm and synchronize our movements to it, which humans find rewarding [8,9]. Additionally, the mere exposure effect finds that repeated exposure to a song can increase enjoyment of it [10], which in turn may enhance groove ratings, as enjoyment correlates with perceived groove [6]. Enjoyment may, in turn, increase listening frequency (i.e., familiarity), creating a positive feedback loop that further amplifies the groove experience.

## Nostalgia as a Groove-Inducing Factor

Unlike familiarity, nostalgia's effect on groove is unclear. Nostalgia involves a yearning for the past [11], and it is most strongly felt for memories formed during late adolescence and early adulthood [12,13]. This phenomenon is attributed to the 'reminiscence bump', in which memories from these formative years are encoded with greater intensity [14]. Despite inducing feelings of loss or pain, nostalgia remains a predominantly positive emotion [15]. Nostalgia activates reward-related areas [16] that are also implicated in groove perception [17]. Sensory inputs, particularly music, are primary triggers for nostalgia [13,18–20]. Research suggests that music-induced nostalgia is stronger when a song is more familiar [13], however, nostalgia and familiarity are distinct, with nostalgia sometimes triggered by elements reminiscent of the past, even if unfamiliar.

## Movement types and groove experience

While studies have examined various factors that affect the urge to move, little work has systematically examined whether groove ratings are affected by considering specific types of movements. Movements like tapping your foot or hand along to the beat, generally moving along, or dancing, have been used to assess groove and are generally treated as equivalent metrics [21]. However, wanting to tap to a song may not be the same experience as wanting to dance, and asking about specific types of movements can affect groove ratings [22]. This distinction may be important as different musical characteristics can influence what parts of the body we move and can elicit different movement characteristics [23,24]. In general, emotions can influence bodily movements in unique spatial, temporal, and energetic ways [25] and this is also true of music-induced emotion [26]. The emotion attached to nostalgia may affect the type of movement one chooses to engage in. The urge to move to music is also related to rhythm [7] and rhythm perception involves interactions between the auditory and motor systems, including the premotor cortex, supplementary motor area, and basal ganglia [27–30], which can be activated even without actual movement. The experience of groove may be considered a precursor to music-induced movement, as high-groove stimuli have been shown to elicit spontaneous rhythmic movements [2]. By analyzing different movement types in relation to groove, we can investigate if these different movements index the same experience, or if they differ.

The present study explored whether groove ratings are affected by nostalgia and whether previously observed differences across movement descriptors (i.e., tap, move, dance) could be reproduced. Specifically, we assessed ratings related to the desire to tap, move, and dance. We hypothesized that 1) nostalgia would significantly predict groove ratings 2) groove ratings would differ across movement descriptors, with differences between tapping, moving, and dancing.

## Materials and methods

### Participants

Participants were recruited using the Prolific Research database [31], and ranged in age from 23 to 28 years old. This age range was selected to ensure a defined participant group with a high likelihood of familiarity with the selected songs, as well as to align the nostalgic songs with participants' reminiscence bump period, when music-evoked nostalgia is particularly strong [13,14]. Additionally, participants were required to either be born in North America or relocated there prior to age five to ensure they would be culturally familiar with the selected songs. Self-reported English fluency was also a requirement, and individuals with self-reported hearing impairments were excluded. For the data analysis, 102 participants were included ($M = 26$ years, $SD = 1.8$ years). This sample size was chosen to align with prior groove studies [17,32] and to provide sufficient power to detect medium-sized effects. Initially, 104 individuals signed up, but two did not complete the study. All participants provided informed written consent before participating. After completing the study, all participants received financial compensation. The non-medical research ethics board at Western University reviewed and approved the experiment.

### Stimuli

The music stimuli were chosen based on a piloting procedure. The following were selected: the top ten songs from the Billboard charts during the participants' adolescent period (2010–2015) and the top ten songs from more recent years (2018–2021), at the time of data collection (August 14–24, 2021). These release periods were selected based on the targeted age of the participants to try to create lists of nostalgic and non-nostalgic songs, respectively. Pilot participants ($N = 4$) in the targeted age range of 23–28 years rated these 100 songs for familiarity and nostalgia. Based off the ratings, 20 early release (nostalgic) songs and 20 late release (non-nostalgic) songs were selected. Both lists had similar overall familiarity ratings. Due to a technical error one of the non-nostalgic songs was played twice, resulting in ratings for only 19 of those songs. All music clips were 25 seconds in length and included the chorus to maximize recognizability. Participants evaluated these forty music excerpts using three 100-point scales gauging their desire to tap, move, and dance, as well as their liking, familiarity, and nostalgia (full song lists and averages for each rating can be found in S1 and S2 Tables).

### Procedure

Participants completed a 45-minute Qualtrics survey consisting of a demographics evaluation, music/dance background assessment, song listening and rating, and debriefing. Within the demographics section, participants reported their age, gender, education level, and ethnicity. The music/dance background section queried the time dedicated to music listening, preferences for music conducive to movement, the completion of the Goldsmiths Dance Sophistication Index (DSI) [31], and the training subscale of The Goldsmiths Musical Sophistication Index (MSI) [33]. The DSI, comprising 26 items, employs a 7-point Likert scale to assess participatory dance experience (20 items) and observational dance experience (6 items), with the former being further separated into four factors: body awareness (6 items), social dancing (6 items), urge to dance (5 items), and dance training (3 items). In the end, only the urge to dance and dance training subscales were used from the DSI.

Within the listening and ratings section, participants were first given a sample music clip and asked to put on headphones and set the volume to a comfortable level. Then the descriptions of what participants were rating (desire to tap, desire to move, desire to dance, enjoyment, familiarity, and nostalgia), were provided along with an example rating scale

(see S3 Table). Each scale ranged from 0–100. The definition given for nostalgia was 'a sentimental longing or wistful affection for the past' [34]. Afterward, the main task started and participants listened to and rated the 40 music clips in randomized order. Midway through the task, an attention check was included, in which participants were presented with only rating scales and asked to set all values to 100. After all ratings were completed, participants were given an open text box to write anything they thought would be important for the researchers to know (e.g., any technical issues) and were provided with a debriefing form.

## Data analysis

Statistical analyses were conducted in Jeffrey's Amazing Statistics Program (JASP Team, 2024). For each participant, averages were calculated for the three movement rating types (desire to tap, desire to move, desire to dance), liking, familiarity and nostalgia for the early and late songs. To assess the song type manipulation on the non-groove ratings a 3 (rating type: liking vs. familiarity vs. nostalgia) x 2 (release period: early vs. late) repeated measures ANOVA on the average values for each rating type was conducted. To determine if groove ratings differed across movement type and release period a 3 (movement rating type: desire to tap vs desire to move vs desire to dance) x 2 (release period: early vs late) repeated measures ANOVA on the average values for each rating type was used. To compare the size of the differences between early and late release periods for each rating type, difference scores were calculated for each of the six rating types, by subtracting the late release period average for each rating type from the early release period average. Two one-way repeated measures ANOVA on the difference scores were used to compare 1) the three non-groove ratings and 2) the three groove ratings. When necessary, Bonferroni-Holm post-hoc tests were completed. Additionally, if the assumption of sphericity was violated then Greenhouse-Geiser corrections were used.

Pearson correlations indicated associations between the six rating scales. Follow-up multiple linear regressions were conducted with the three movement rating types as outcome variables, with familiarity and nostalgia as potential predictors. Liking was not included as a predictor because pleasure is an inherent part of the groove experience, as it is the 'pleasurable urge to move'. Groove and liking are often highly correlated [2,3,35] and including it as a predictor would likely dilute other effects. Additionally, Pearson correlations between the three movement rating types, the music training subscale, the dance training subscale, and the urge to dance subscale assessed relationships between music and dance experience and the different groove ratings.

The above analyses were conducted on participants' average ratings across songs to understand the general relationships between the three movement types, nostalgia, and familiarity. To further explore the strength of these relationships, Pearson correlation values were also calculated *within* each participant for the ratings of the three movement types with nostalgia and familiarity across songs. The correlation coefficients for each of the six correlations (i.e., tap-nostalgia, move-nostalgia, dance-nostalgia, tap-familiarity, move-familiarity, dance-familiarity) for each participant were put into a 3 (movement rating type: desire to move vs desire to tap vs desire to dance) x 2 (familiarity vs nostalgia) repeated measures ANOVA on the individual correlation coefficients for the six correlations with Bonferroni-Holm post-hoc tests. Three participants were excluded from this analysis as they had constant ratings for one of the rating categories and therefore correlation coefficients could not be calculated.

## Results

To assess our release period manipulation, and whether the ratings differed across song types, the three non-groove ratings (i.e., liking, familiarity, and nostalgia) for the two release periods (early versus late) were compared. As shown in Fig 1, there was a large effect of non-groove rating types, $F(1.9, 194.9) = 186.48$, $p < .001$, $\eta_p^2 = .65$, and a large effect of release period $F(1, 101) = 165.61$, $p < .001$, $\eta_p^2 = .62$. Post hoc tests indicated early songs ($M = 78.1$) were rated higher than late songs ($M = 61.1$, $p < .001$) on all three ratings. These main effects are qualified by a significant interaction, with a medium effect size, between non-groove factors and release period, $F(2, 202) = 139.24$, $p < .001$, $\eta_p^2 = .08$. Further, the

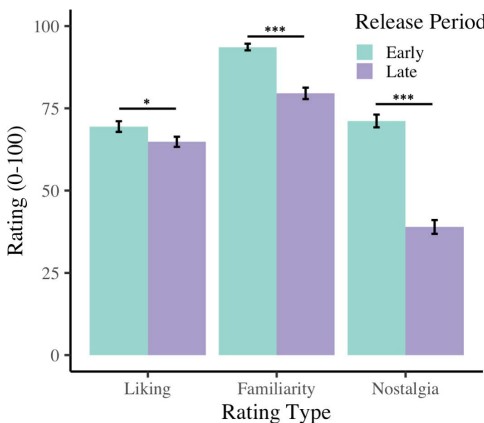

**Fig 1. Ratings of liking, familiarity, and nostalgia for music released during participants' adolescent period (early) or more recently (late).**

three non-groove ratings' differences between early and late averages were significantly different with a large effect size, $F(1.8, 202) = 139.24$, $p < .001$, $\eta_p^2 = .58$. Post hoc tests revealed that the difference between the early and late averages were lowest for liking ($M = 4.6$), followed by familiarity ($M = 14.1$), and nostalgia ($M = 32.2$) had the largest difference, $p$'s $< .001$.

Fig 2 shows the ratings of different movement types for the early and late release periods. There was a large effect of movement type, $F(1.57, 158.5) = 123.99$, $p < .001$, $\eta_p^2 = .55$, and a large effect of release period $F(1, 101) = 65.01$, $p < .001$, $\eta_p^2 = .39$. Post hoc tests indicated that tap ratings ($M = 67.4$) were significantly higher than move ratings ($M = 61.1$, $p < .001$) and dance ratings ($M = 49.7$, $p < .001$). Move ratings were also higher than dance ratings ($p < .001$). Early songs ($M = 63.9$) were rated higher than late songs ($M = 54.8$, $p < .001$). These main effects are qualified by a small interaction between groove and release period, $F(1.44, 202) = 4.49$, $p = .024$, $\eta_p^2 = .04$. The three groove ratings' differences between early and late averages were significantly different with a small effect size, $F(1.44, 145.55) = 4.491$, $p = .023$, $\eta_p^2 = .043$. Post hoc tests revealed that the difference between dance ratings ($M = 10.4$) was smaller than the difference between tap ratings ($M = 8.3$, $p = .01$), but neither differed from the difference between move ratings ($M = 8.9$, $p$'s $> .1$).

Fig 3A shows the Pearson's correlations between ratings of desire to tap, desire to move, desire to dance, liking, familiarity, and nostalgia. All three movement rating types, liking, nostalgia, and familiarity were strongly correlated, except dancing and familiarity. Separate multiple linear regressions were carried out for the three movement rating types as the outcome variables, with familiarity and nostalgia as predictors. The model was significant for desire to tap ratings $F(2, 99) = 17.17$, $p < .001$, $R^2 = .24$, with both familiarity ($t = 3.34$, $p = .001$), and nostalgia ($t = 3.45$, $p < .001$) as significant predictors. Desire to move was also significantly predicted by the model $F(2, 99) = 11.54$, $p < .001$, $R^2 = .17$, with familiarity ($t = 2.57$, $p = .012$) and nostalgia ($t = 2.99$, $p = .004$) as significant predictors. Lastly, the model for desire to dance ratings was significant $F(2, 99) = 11.14$, $p < .001$, $R^2 = .17$, but nostalgia was a significant predictor ($t = 4.18$, $p < .001$) while familiarity ($t = 0.70$, $p = .49$) was not.

We also correlated each participant's individual song ratings for the three movement types with nostalgia and familiarity. Fig 3B shows the differences across the correlation coefficients for the different movement types and nostalgia and familiarity. There was a large effect of movement rating type, $F(2, 98) = 19.89$, $p < .001$, $\eta_p^2 = .17$, and a large effect of familiarity or nostalgia rating type, $F(1, 98) = 15.51$, $p < .001$, $\eta_p^2 = .14$. These main effects are qualified by a significant interaction, with a large effect size, between movement rating type and familiarity or nostalgia rating type, $F(2, 98) = 18.18$, $p < .001$, $\eta_p^2 = .16$. There was no significant difference between familiarity and nostalgia correlations with desire to tap ($p = .08$). Nostalgia and desire to move were more correlated than familiarity and desire to move ($p = .04$). Similarly, nostalgia and desire to dance ($p < .001$) were more correlated than familiarity and desire to move.

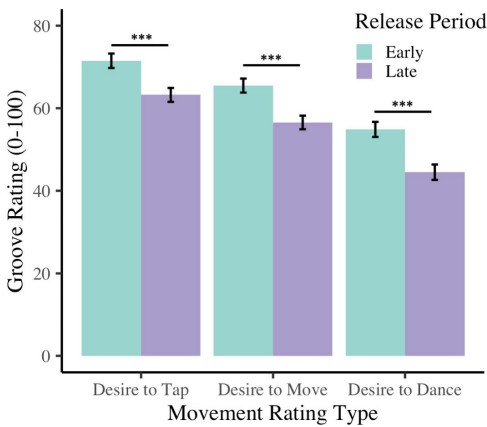

**Fig 2. Groove movement rating types (tap, move, and dance) for music released during participants' adolescent period (early) or more recently (late).**

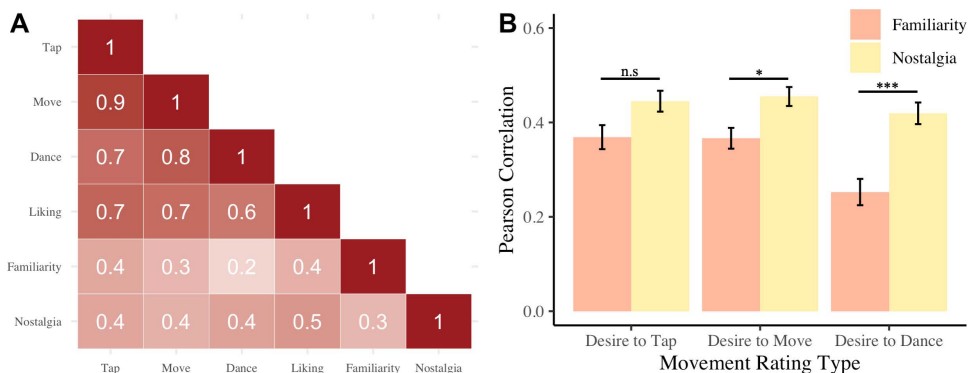

**Fig 3. Pearson Correlation for each movement rating type (tap, move, and dance) with familiarity and nostalgia.** All correlation coefficient were significant at $p < .001$, except the dance-familiarity correlation.

Table 1 shows the Pearson's correlations between participant's average ratings of desire to tap, desire to move, desire to dance, MSI training ($M = 21$, $SD = 9.7$), DSI urge to dance ($M = 22.3$, $SD = 5.4$), and DSI dance training ($M = 6.7$, $SD = 4.4$). Desire to tap did not correlate with MSI, DSI urge to dance, or DSI dance training. Desire to move and desire to dance correlated positively with both DSI urge to dance and DSI dance training scores. The DSI urge to dance and DSI dance training scores were also significantly correlated.

## Discussion

The current study examines the relationship between nostalgia and groove, by having participants listen to and rate songs from different periods of their lives. The two song types were defined by early versus late release periods, early release period songs were popular during the participants adolescence, and late release period songs were more recent popular songs. The early release period (i.e., older songs) received higher ratings of liking, familiarity, and nostalgia than the late release period. However, the difference in ratings across release period was largest for nostalgia. Across our three groove movements - tapping, moving, and dancing – the early release period songs had higher groove ratings for all three movement types than the late release period songs. Overall, the early release period songs from the participants' "reminiscence

**Table 1. Pearson's correlation matrix for desire to tap, move, dance, music training, urge to dance, and dance training.**

|  | Desire to tap | Desire to move | Desire to dance | Music Training | Urge to dance | Dance training |
|---|---|---|---|---|---|---|
| Desire to tap | — |  |  |  |  |  |
| Desire to move | 0.87*** | — |  |  |  |  |
| Desire to dance | 0.67*** | 0.75*** | — |  |  |  |
| MSI: Training | −0.054 | −0.082 | −0.11 | — |  |  |
| DSI: Urge to dance | 0.15 | 0.22* | 0.38*** | 0.067 | — |  |
| DSI: Dance training | 0.099 | 0.21* | 0.26** | 0.18 | 0.42*** | — |

* p < .05, ** p < .01, *** p < .001.

bump" period, were, more familiar, more liked, and made the participants want to tap, move, and dance more, compared to newer songs.

Based on the above analyses, it is not possible to attribute the higher groove ratings solely to nostalgia, as familiarity and liking were also higher for the early release period songs. It may be that the nostalgic songs affect groove because they are familiar and well liked. To separate these factors, linear regression analyses were conducted to compare familiarity and nostalgia's contributions to the ratings of the different movement types across all songs. Familiarity and nostalgia were each significant individual predictors of move and tap ratings, but only nostalgia was a significant predictor of dance ratings. Therefore, nostalgia is associated with groove ratings over and above the effect of familiarity, suggesting its influence is not solely due to familiarity. Similarly, nostalgia predicts how much people want to move, and especially, dance.

The previous analyses averaged ratings across songs, allowing for general conclusions about the relationships between the ratings, such as ratings of higher nostalgia overall related to higher desire to dance. Additional analyses compared the *strength* of the relationships (correlation coefficients) between the different movement ratings and the nostalgia and familiarity ratings at a participant-by-participant level. Each participants' correlations were calculated across ratings of all songs, and then coefficients for the different movement types were compared. The relationship between nostalgia and the three movement types was stronger than their relationship with familiarity, especially for the desire to dance. The more nostalgic a song was, the stronger the reported desire to tap, move, and dance, whereas the more familiar a song was, the stronger the reported desire to only tap and move. Therefore, it seems that in addition to nostalgic music eliciting groove, it is specifically important for making people want to dance.

The current study also supports the notion that not all groove related movements are rated equally [22]. The three movement types, which in previous literature has rarely been systematically compared, yielded different groove ratings and different relationships with variables of interest. Notably, nostalgia had a unique relationship with the desire to dance along to music. Several analyses indicated that dancing was rated differently from wanting to tap or move to music. First, there was no significant association between familiarity and the desire to dance once nostalgia was accounted for. One might expect an association, based on previous research [1,2,6,7]. Second, the desire to dance ratings were lower overall than the tap and move ratings. Moving along to music and dancing along to music seem very similar, but with dance there may be additional social factors, such as an aversion to or anxiety about dancing that results in lower ratings. The relationship between nostalgia and dance is especially interesting, as anecdotally people enjoy dancing to nostalgic songs.

One potential mediating factor between nostalgia and groove is arousal. Arousal is associated with music-evoked nostalgia [36] and in the theoretical model of groove [37], arousal is proposed to contribute to the urge to move. It follows that nostalgic music may elicit arousal, and arousal contributes to the urge to move, resulting in higher ratings. The role of arousal is also supported by the movement specific findings. It may be that wanting to dance along to music compared to

wanting to tap or move, requires more arousal, which nostalgic music provides. However, as arousal was not measured here, interactions between groove ratings and arousal cannot be explored by the current study.

We also examined music and dance training effects, as experience with both music and dance could affect participants' perception of groove. Previously, there has been mixed findings on whether music training affects groove perception [6,7,32]. Studies that have found effects with music training have often examined specific features of music or less popular genres [38]. Popular music songs were used in this study, which may be why no effects of music training were found.

Only a few studies have investigated the role of dance training on groove perception. In our sample, dance training positively correlated with the desire to move and dance. Dance training involves the coupling of movements and music, and this experience may result in a stronger desire to move when listening to music. Further, people who have participated in dance training may have less anxiety around dancing, which may have resulted in higher ratings. The lack of correlation between dance training and wanting to tap also highlights how the different movements associated with groove are not completely comparable. Additionally, participants' score on the DSI urge to dance subscale was also only related to their desire to move and dance but not to tap along. This is likely due to the items in the urge to dance subscale of the DSI being specific to dancing and not general movements/tapping. However, this also suggests a difference between wanting to tap versus wanting to dance to music, which is typically not differentiated in studies of the concept of groove.

## Limitations and future directions

While this study provides insight into the relationship between nostalgia and groove, several limitations could be considered. First, the study focused exclusively on young adults (23–28 years old), limiting the generalizability of the findings to older populations. In particular, older populations would be of interest to study, as the sensation of nostalgia generally increases over the lifespan [39]. Future research could investigate whether different age groups exhibit varying movement responses to nostalgic music, as understanding the properties of music that enhance movement could inform applications of music to gait rehabilitation [40]. An additional limitation is that, while familiarity and nostalgia were measured separately, their measurement at similar points in time may have influenced ratings, making it difficult to fully disentangle their individual effects. Despite efforts to control for familiarity, the newer songs were still rated as less familiar than the older ones. As a result, additional analyses were necessary to account for familiarity, but the current study design cannot fully isolate nostalgia from familiarity. The study was also conducted remotely, meaning listening conditions could vary across participants and were not under experimental control, potentially affecting responses. Finally, as this study did not comprehensively quantify musical features across stimuli (e.g., differences in syncopation, micro timing, tempo range, etc.), the songs may differ in other characteristics that contribute to groove.

Further research is needed to determine whether nostalgia alone drives the observed effects, or if differences in familiarity, listening conditions, musical properties, and age contribute. Future studies could isolate nostalgia's unique relationship to groove by controlling for these factors, as well as examining nostalgia's specific role in the urge to dance across different populations.

## Conclusions

Using nostalgic songs to get people out onto the dance floor or elicit strong reactions is common practice. This is the first study to empirically investigate whether nostalgia relates to people's urge to move along to music. The results here suggest that nostalgia is associated with higher groove ratings, beyond the effects of familiarity. Nostalgia also seems most powerful in eliciting the desire to dance. Further, the three different movement ratings yielded different levels of groove overall and different relationships with nostalgia and dance training. The differences found between the three movement types suggests that wanting to dance to music is different than wanting to tap along. Lastly, there were no effects of music training, but there were effects of dance training, with dancers experiencing the desire to move and dance more than non-dancers. This suggests that individuals with dance training tend to report a stronger desire to move or dance to music.

## Supporting information

**S1 Table. Average ratings and song information for early songs (2010–2015).**
(PDF)

**S2 Table. Average ratings and song information for late songs (2018–2021).**
(PDF)

**S3 Table. Definitions and instructions for each rating scale used in the study.** *Note.* Participants were told that for items a–c, "Note that we are not asking you how much the music made you tap, move, or dance along, but how much you had a perceived feeling of wanting to tap, move, or dance to the music.".
(PDF)

## Author contributions

**Conceptualization:** Riya K. Sidhu, Jessica A. Grahn.

**Data curation:** Riya K. Sidhu, Hong Zheng.

**Formal analysis:** Riya K. Sidhu, Diana M. Urian, Hong Zheng.

**Funding acquisition:** Jessica A. Grahn.

**Investigation:** Riya K. Sidhu.

**Methodology:** Riya K. Sidhu, Hong Zheng.

**Project administration:** Riya K. Sidhu, Hong Zheng.

**Supervision:** Riya K. Sidhu, Jessica A. Grahn.

**Validation:** Riya K. Sidhu.

**Visualization:** Riya K. Sidhu, Diana M. Urian.

**Writing – original draft:** Riya K. Sidhu, Diana M. Urian.

**Writing – review & editing:** Riya K. Sidhu, Diana M. Urian, Jessica A. Grahn.

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
