## [Decision Letter · Decision Letter 0]

10 Mar 2025

PONE-D-25-03512Throwbacks That Move Us: The Dance-Inducing Power of NostalgiaPLOS ONE

Dear Dr. Sidhu,

Thank you for submitting your manuscript to PLOS ONE. After careful consideration, we feel that it has merit but does not fully meet PLOS ONE’s publication criteria as it currently stands. Therefore, we invite you to submit a revised version of the manuscript that addresses the points raised during the review process.

Please take reviewers' points constructively and respectfully address them including the critical ones. As an Academic Editor, I believe all the points that reviewers raised are valid and worth discussing.

We look forward to receiving your revised manuscript.

Kind regards,

Seung-Goo Kim, Ph.D.

Academic Editor

PLOS ONE

3. Please include a separate caption for each figure in your manuscript

Additional Editor Comments (if provided):

Reviewers' comments:

Reviewer's Responses to Questions

**Comments to the Author**

1. Is the manuscript technically sound, and do the data support the conclusions?

Reviewer #1: Yes

Reviewer #2: Yes

Reviewer #3: No

2. Has the statistical analysis been performed appropriately and rigorously? 

Reviewer #1: Yes

Reviewer #2: No

Reviewer #3: Yes

3. Have the authors made all data underlying the findings in their manuscript fully available?

Reviewer #1: Yes

Reviewer #2: Yes

Reviewer #3: Yes

4. Is the manuscript presented in an intelligible fashion and written in standard English?

Reviewer #1: Yes

Reviewer #2: Yes

Reviewer #3: Yes

5. Review Comments to the Author

Reviewer #1: Thank you for your manuscript reporting on an interesting piece of research. Please note the following points for review:

1. The manuscript title does not have “music” as part of it. Including “music” within the title would avoid confusion and give a clearer idea of the focus of the manuscript & research.

2. The choice of only ages under 30 for participants needs to be justified further. With regard to nostalgia, this is significantly more relevant for older people. Whilst I understand the reasons for only including under 30s in this research, the issue of age does need to be addressed in more detail.

3. In the conclusion, more needs to be added about the limitations of the research (including as per point 2 above).

4. More could also be added about potential implications of your research - how can a greater understanding of the dance-inducing qualities of nostalgic music improve how we treat those with physical movement problems, for example?

Reviewer #2: This is a very interesting research article exploring the desire to dance, tap, or move for nostalgic and familiar songs. Furthermore, this investigates the role of nostalgia among young persons and their desire to move. However, the relatively short article can be expanded with some further information to bestow clarity to the reader.

In the Procedure section (Lines : 122 - 133), the manner in which the stimuli were presented is not immediately clear. It is imparative that the reader understand the way the experiment was conducted. I would include a block diagram with the user interface presented to the participant explaining the flow of the experiment. Were the stimuli randomized to avoid order biases in participants ? This information is crucial and should be included.

The data analysis conducted on mean ratings is outlined well. You are using a parametric analysis which is understandable with the large sample size used. Partial eta squared values are reported as effect sizes (Excellent!). However, this should be accompanied by a wording of how big the effect is (e.g. Large effect, Medium effect and so on.)

Lines 184 - 195, the author mentions post-hoc test. Which post-hoc tests were conducted ? students t-test. There are multiple comparisons yet no mention of p-value corrections. This information needs to be included.

If t-tests were used, were degrees of freedom adjusted if variances were different ?

Overall, I believe this work will see marked improvements if the above information is provided. For this reason, I am willing to accept pending "Major" revisions.

P.S* The attention check which requires the participant to rate everything as 100 should be included throughout the experiement at random rather than once halfway! This ensures sustained engagement over the full course of the experiment.

Reviewer #3: The authors present an interesting topic, but the study design has major weaknesses that must be resolved. In its current state, the findings are confounded by many variables, making it difficult to draw meaningful conclusions. Many of these issues can be addressed with a robust study design and methodological improvements. Better control for the confounding factors is necessary. As they also acknowledge in the discussion, the data suggests that the groove ratings are clearly influenced by familiarity and liking. Only for the dance, nostalgia but not familiarity predicted how much people wanted to dance. In addition to familiarity, various musical features that are known to influence groove perception (e.g., syncopation, micro timing, tempo range, attack onsets), none of which are mentioned nor further explored in the manuscript. Are the early and late releases comparable or do they differ in terms of their musical features that are generally known to affect groove perception?

Secondly, in several places in the manuscript, the authors draw much larger conclusions than their findings suggest. This is misleading to the reader. For example, findings specific to desire to dance are attributed to the link between nostalgia and groove as a whole. In various places, causality is inferred from correlations. Again, findings related to dance training are interpreted in a way that the authors draw wider conclusions than the data indicates.

Third, several important methodological issues are not addressed. The hypotheses are vague and the analysis plan does not fully follow the hypotheses laid earlier in the manuscript. There was no mention of employing any technique to control for sequence effects. Did you take any measures to account for these? Decisions related to sampling were also not clear. Did you do any a-priori power analysis to determine the sample size? What was the stopping rule?

On a separate note, I selected "Yes" to the data availability question above (Q3), but the link to the data on OSF is not provided. Hence the data were not available at the time of review. The authors should either provide the link or mention in the additional data availability information section that the URLs/accession numbers/DOIs will be available only after acceptance of the manuscript for publication.

6. PLOS authors have the option to publish the peer review history of their article (what does this mean? ). If published, this will include your full peer review and any attached files.

**Do you want your identity to be public for this peer review?** For information about this choice, including consent withdrawal, please see our Privacy Policy .

Reviewer #1: No

Reviewer #2: **Yes: ** Aravindan Joseph Benjamin

Reviewer #3: No

---

## [Author Response · Author response to Decision Letter 1]

23 Mar 2025

Reviewer #1:

1. The manuscript title does not have “music” as part of it. Including “music” within the title would avoid confusion and give a clearer idea of the focus of the manuscript & research.

The title has been changed to “Throwbacks That Move Us: The Dance-Inducing Power of Nostalgic Songs” to make it clearer. Thank you for this note.

2. The choice of only ages under 30 for participants needs to be justified further. With regard to nostalgia, this is significantly more relevant for older people. Whilst I understand the reasons for only including under 30s in this research, the issue of age does need to be addressed in more detail.

We knew we needed to have a defined age group, and this age group was selected so we could also collect data from the university subject pool (as another possible recruitment strategy that we ended up not using). Past that, at the time of the research, the primary author was in that age range and had more familiarity with the songs chosen, which helped in stimulus selection (e.g., knowing if a song was slow or inappropriate).

In the participants section (Page 4, Line 89-91), we have changed the wording to add justification. Replicating this study in different age ranges would be interesting because, as you mention, nostalgia is more relevant for older people, and this has been noted in the discussion (Page 13, Lines 315-317).

3. In the conclusion, more needs to be added about the limitations of the research (including as per point 2 above).

Thank you for this suggestion. We have now included a dedicated Limitations and Future Directions section (Page 13, Lines 312-329) to clearly outline the constraints of our study. This section highlights the age restriction of our sample, the challenge of fully disentangling familiarity from nostalgia, the remote testing environment, and the lack of direct control over musical features that influence groove. Additionally, we now explicitly acknowledge the need for future research to isolate nostalgia’s unique relationship with groove by controlling for these factors. We feel that these additions provide a clearer discussion of the study’s limitations and directions for further investigation.

4. More could also be added about potential implications of your research - how can a greater understanding of the dance-inducing qualities of nostalgic music improve how we treat those with physical movement problems, for example?

In response, we have expanded our discussion on the potential applications of nostalgia-induced movement motivation. Specifically, we have linked our findings to gait rehabilitation, referencing work by Leow et al. (2014)(Page 13, line 317-319). We now suggest that understanding the properties of music that enhance movement—particularly nostalgic music—could inform interventions aimed at improving motor function in clinical populations, like other research that is done in our lab.

Reviewer #2:

1. In the Procedure section (Lines : 122 - 133), the manner in which the stimuli were presented is not immediately clear. It is imparative that the reader understand the way the experiment was conducted. I would include a block diagram with the user interface presented to the participant explaining the flow of the experiment. Were the stimuli randomized to avoid order biases in participants ? This information is crucial and should be included.

We have edited that paragraph to make the stimulus presentation clearer (Page 6, Line 128-138). We have also added an example rating scale in the supplemental materials to aid in understanding of the procedure. If it is still confusing, we can consider adding a block diagram in addition.

Yes, the stimuli were randomized, and this is also noted in the procedure (Page 6, Line 134).

2. The data analysis conducted on mean ratings is outlined well. You are using a parametric analysis which is understandable with the large sample size used. Partial eta squared values are reported as effect sizes (Excellent!). However, this should be accompanied by a wording of how big the effect is (e.g. Large effect, Medium effect and so on.)

Thank you for this note, I have added in the wording when describing the effects from the ANOVA throughout the Results section

3. Lines 184 - 195, the author mentions post-hoc test. Which post-hoc tests were conducted ? students t-test. There are multiple comparisons yet no mention of p-value corrections. This information needs to be included.

If t-tests were used, were degrees of freedom adjusted if variances were different ?

For our Bonferroni post-hoc tests, we applied the Holm correction to adjust for multiple comparisons. This information has now been included in the data analysis section (Page 7, Line 153).

Yes, if sphericity was violated, then Greenhouse-Geiser corrections were used, this is also now specified in the Data Analysis section (Page 8, Line 153-154) and I have double checked and updated values to be the corrected ones.

P.S* The attention check which requires the participant to rate everything as 100 should be included throughout the experiement at random rather than once halfway! This ensures sustained engagement over the full course of the experiment.

This is a good note for future studies, thank you.

Reviewer #3:

Better control for the confounding factors is necessary. As they also acknowledge in the discussion, the data suggests that the groove ratings are clearly influenced by familiarity and liking. Only for the dance, nostalgia but not familiarity predicted how much people wanted to dance. In addition to familiarity, various musical features that are known to influence groove perception (e.g., syncopation, micro timing, tempo range, attack onsets), none of which are mentioned nor further explored in the manuscript. Are the early and late releases comparable or do they differ in terms of their musical features that are generally known to affect groove perception?

We appreciate this observation and acknowledge that musical features known to influence groove perception were not explicitly analyzed in this study. While we selected songs based on their release period and pilot ratings, we did not control for characteristics such as tempo, syncopation, or microtiming, which may have contributed to groove ratings independently of nostalgia. As you mentioned, there are also several potential confounds such as familiarity, and these are also noted in the Limitations section (Page 13, Line 323-330).

Secondly, in several places in the manuscript, the authors draw much larger conclusions than their findings suggest. This is misleading to the reader. For example, findings specific to desire to dance are attributed to the link between nostalgia and groove as a whole. In various places, causality is inferred from correlations. Again, findings related to dance training are interpreted in a way that the authors draw wider conclusions than the data indicates.

Thank you for highlighting this concern. We have carefully reviewed the manuscript and identified instances where causal language was used. We have revised these to better reflect the correlational nature of our findings. Specifically, we have replaced phrasing that implied direct effects of nostalgia on groove with wording that more accurately conveys associations. Below are the key changes made, with their corresponding page and line numbers:

1. Page 10, Line 244 – Changed "the effect of nostalgia on groove" → "the relationship between nostalgia and groove" to avoid implying causality.

2. Page 11, Line 261-262 – Changed "nostalgia affects groove ratings over and above the effect of familiarity" → "nostalgia is associated with groove ratings over and above the effect of familiarity."

3. Page 11, Line 271-273– Changed "The more nostalgic a song was, the more it made people want to tap, move, and dance" → "The more nostalgic a song was, the stronger the reported desire to tap, move, and dance."

4. Page 14, Lines 339-395 – Changed "The results here suggest that nostalgia does increase groove, beyond the effects of familiarity" → "The results here suggest that nostalgia is associated with higher groove ratings, beyond the effects of familiarity."

These revisions ensure that our language appropriately reflects the correlational nature of our study, aligning with the methodological constraints of the design. We appreciate this feedback, as it has helped us clarify our interpretations for the reader.

We have carefully reviewed the manuscript and have softened our interpretation of the dance training findings to ensure they are not framed as causal. Below are the key changes made, with corresponding page and line numbers:

1. Page 14, Line 346-347 – Changed "This suggests that dance training, which results in experience with coupling music and movement, may increase the desire to move or dance to music."

→ "This suggests that individuals with dance training tend to report a stronger desire to move or dance to music."

Third, several important methodological issues are not addressed. The hypotheses are vague and the analysis plan does not fully follow the hypotheses laid earlier in the manuscript.

Thank you for this feedback. We have revised the Introduction (Page 4, Lines 81-85) to explicitly clarify our hypotheses. The revised text now reads:

"The present study explored whether groove ratings are affected by nostalgia and whether previously observed differences across movement descriptors (i.e., tap, move, dance) could be reproduced. Specifically, we assessed ratings related to the desire to tap, move, and dance. We hypothesized that (1) nostalgia would significantly predict groove ratings and (2) groove ratings would differ across movement descriptors, with differences between tapping, moving, and dancing."

This revision ensures that our hypotheses are clearly stated and aligns them more explicitly with our analytical approach.

There was no mention of employing any technique to control for sequence effects. Did you take any measures to account for these?

Yes, the order of song presentation was randomized for each participant to prevent order effects from influencing ratings. This information is included in the Methods section on page 6, line 134:

"Afterward, the main task started and participants listened to and rated the 40 music clips in randomized order."

This ensures that potential confounds related to presentation order do not impact the study’s findings.

Decisions related to sampling were also not clear. Did you do any a-priori power analysis to determine the sample size? What was the stopping rule?

Thank you for your observation regarding our sample size. We acknowledge the importance of selecting an appropriate sample size to ensure the robustness and generalizability of research findings. Our decision to include 102 participants was informed by several considerations:

1. Consistency with Existing Research: Sample sizes in groove perception studies vary depending on the research design and objectives. For example:

o Frühauf et al. (2013) investigated the influence of microtiming on perceived groove quality using 93 participants.

o Matthews et al. (2020) examined the correlates of musical groove perception with 124 participants.

Our sample size of 102 participants aligns with these studies, suggesting it is within the typical range for research in this field.

2. Resource Constraints: This study was conducted as part of an undergraduate project, which limited the resources and time available for data collection. We aimed to balance these constraints while still obtaining meaningful and reliable results.

3. Statistical Power Considerations: We aimed to detect medium effect sizes with adequate statistical power. Based on conventional power analysis guidelines, a sample size of approximately 100 participants is typically sufficient to detect medium effect sizes with a power of 0.80 and an alpha level of 0.05.

We have incorporated these considerations into the Participants section of the manuscript on page 5, line 96-97:

"This sample size was chosen to align with prior groove studies (Frühauf et al., 2013; Matthews et al., 2020) and to provide sufficient power to detect medium-sized effects."

On a separate note, I selected "Yes" to the data availability question above (Q3), but the link to the data on OSF is not provided. Hence the data were not available at the time of review. The authors should either provide the link or mention in the additional data availability information section that the URLs/accession numbers/DOIs will be available only after acceptance of the manuscript for publication.

There is now a link with the data available.

---

## [Decision Letter · Decision Letter 1]

1 Apr 2025

Throwbacks That Move Us: The Dance-Inducing Power of Nostalgic Songs

PONE-D-25-03512R1

Dear Dr. Sidhu,

We’re pleased to inform you that your manuscript has been judged scientifically suitable for publication and will be formally accepted for publication once it meets all outstanding technical requirements.

Kind regards,

Seung-Goo Kim, Ph.D.

Academic Editor

PLOS ONE

Additional Editor Comments (optional):

Reviewers' comments:

Reviewer's Responses to Questions

**Comments to the Author**

1. If the authors have adequately addressed your comments raised in a previous round of review and you feel that this manuscript is now acceptable for publication, you may indicate that here to bypass the “Comments to the Author” section, enter your conflict of interest statement in the “Confidential to Editor” section, and submit your "Accept" recommendation.

Reviewer #1: All comments have been addressed

Reviewer #2: All comments have been addressed

2. Is the manuscript technically sound, and do the data support the conclusions?

Reviewer #1: Yes

Reviewer #2: Yes

3. Has the statistical analysis been performed appropriately and rigorously? 

Reviewer #1: Yes

Reviewer #2: Yes

4. Have the authors made all data underlying the findings in their manuscript fully available?

Reviewer #1: Yes

Reviewer #2: Yes

5. Is the manuscript presented in an intelligible fashion and written in standard English?

Reviewer #1: Yes

Reviewer #2: Yes

6. Review Comments to the Author

Reviewer #1: Thank you for your revisions, addressing the concerns and issues raised.

Just one small grammar fix: line 117 - “demographic’s” should not have an apostophe, i.e. should be “demographics”.

Reviewer #2: The author has applied all of the suggested corrections from me. Therefore, I am satisfied in terms of having this manuscript accepted for publication.

7. PLOS authors have the option to publish the peer review history of their article (what does this mean? ). If published, this will include your full peer review and any attached files.

**Do you want your identity to be public for this peer review?** For information about this choice, including consent withdrawal, please see our Privacy Policy .

Reviewer #1: No

Reviewer #2: **Yes: ** Aravindan Joseph Benjamin

---

## [Editor Report · Acceptance letter]

PONE-D-25-03512R1

PLOS ONE

Dear Dr. Sidhu,

I'm pleased to inform you that your manuscript has been deemed suitable for publication in PLOS ONE. Congratulations! Your manuscript is now being handed over to our production team.

Kind regards,

on behalf of

Dr. Seung-Goo Kim

Academic Editor

PLOS ONE